# Forecasting and Planning Method for Taxi Travel Combining Carbon Emission and Revenue Factors—A Case Study in China

**DOI:** 10.3390/ijerph191811490

**Published:** 2022-09-13

**Authors:** Lixin Yan, Bowen Sheng, Yi He, Shan Lu, Junhua Guo

**Affiliations:** 1School of Transportation Engineering, East China Jiaotong University, Nanchang 330013, China; 2Intelligent Transportation Systems Research Center, Wuhan University of Technology, Wuhan 430063, China; 3Institute of Intelligence Science and Engineering, Shenzhen Polytechnic, Shenzhen 518000, China

**Keywords:** carbon emissions, GPS data, VSP Model, order revenue, comprehensive evaluation, cluster analysis

## Abstract

The efficiency and emission levels of taxi operations are influenced by taxi drivers’ empirical judgments of hotspot travel areas. In this study, we exploited vehicle specific power (VSP) approaches and taxi trajectory data in a 1000 × 1000 m grid to calculate emission and revenue efficiency-related indicators and explored their spatial and temporal characteristics. Then, the entropy weight TOPSIS method was employed to identify the grids with the top comprehensive ranking of the indicators in the period to replace the driver experience. Finally, the k-means clustering method was utilized to identify the recommended road segments in the hotspot grid. The data from Nanchang City in China showed the following. (1) The study area was divided into 7553 grids, and the main travel and emission areas were located in the West Lake, Qingyunpu and Qingshan Lake districts (less than 200 grids). However, revenue efficiency-related indicators in this region are at a moderately low level. For example, the order revenue was about 0.9–1.2 RMB/min, and the average was 1.3–1.5 RMB/min. Areas with high trip demand had low revenue efficiency. (2) Five indicators related to emissions and revenue efficiency were selected. Of these, grid boarding points (G-bp) maintained the highest weight, reaching a maximum of 0.48 from 7:00 a.m. to 9:00 a.m. The ranking of secondary indicators was time varying. Hotspot grids and road segments were identified within each period. For example, from 1:00 a.m. to 3:00 a.m., (66,65), (68,65) were identified as hotspot grids. People’s Park North Gate near the road was identified as the recommended section from 1:00 a.m. to 3:00 a.m. This study can provide recommended grids and sections for idle cruising taxis.

## 1. Introduction

### 1.1. Background

Transportation-related CO_2_ emissions account for approximately 23% of total global emissions [1]. Of these, road transport is the main driver of growth in transport-related carbon emissions, accounting for more than 75% of global transport carbon emissions [2]. Daily travel dominates the carbon emissions associated with road traffic [3,4,5,6]. Taxis, as an important mode of transportation for daily travel, contributes to more emissions at a higher level. Especially in recent years, some studies have confirmed that online car-hailing has lower emission levels than taxis in China. Taxis tends to have longer idle trips [7,8]. The traditional way taxi drivers find passengers largely depends on the experience of the taxi driver [9]. This choice behavior leads to a spatiotemporal mismatch between taxi supply and passenger demand, resulting in a large number of idle taxi emissions. Reducing the taxi idle driving rate requires taxi drivers to have experience and judgment of active road sections. In the past decade, Global Positioning System (GPS) data have become an important tool for investigating urban trip choice behaviors and activities [10]. The analysis of GPS data can replace the empirical judgment of taxi drivers. Emission levels within the region, as well as high-income road segments, are determined from the taxi trip network. Traffic managers can improve traffic configurations in high-emission, low-income areas and limit or promote the number of taxis in certain areas, thereby optimizing the overall traffic layout.

### 1.2. Related Works

We discuss the contribution of GPS data to studies related to travel emissions and travel hotspots from four perspectives.

#### 1.2.1. Calculation Method for Transportation-Related Carbon Emissions

Bottom-up carbon emission measurement methods are widely used in the field of transportation. Emissions are measured primarily using emission factors, transport category and distance traveled [11]. However, the lack of travel time and spatial location makes it difficult to monitor dynamic emissions in the region. Relying on travel surveys in regional emissions calculations, which are considered time consuming and inaccurate [12]. For this purpose, GPS data have received increasing attention since these data accurately reflect the driving state of vehicles [13]. Vehicle Specific Power (VSP) models can calculate emissions using GPS data. The relationship between the average emission rate and the GPS-based driving parameters was established by regression fitting. The VSP model and carbon emission factors allow the analysis of the dynamic spatial and temporal distribution of travel emissions [14]. Luo et al. [15] exploited GPS data to analyze the energy consumption and emissions of taxis in Shanghai and their spatial and temporal distribution. The results showed that there are two central points of emissions, one in the city center and the other in the Hongqiao transportation hub. Zhang et al. [16] adopted the VSP model to analyze the spatial and temporal distribution of carbon emissions from taxis in Beijing. The study showed that emissions were higher on ring roads, highways and large intersections. However, studies lack an exploration of the factors that influence the generation of travel carbon emissions. Acuto et al. [17] through the accuracy test of the vehicle specific power (VSP) model and the integrated vehicle emission calculation model of the micro traffic simulation (AIMSUN), proposed that there is great potential in traffic big data in road environmental performance evaluation. Patiño-Aroca et al. [18] exploited the International Vehicle Emissions Model (IVE) to estimate the emissions caused by fleet travel at a resolution of 1 km × 1 km. The VSP model was used to correct the base emission factor. The results showed that private passenger cars were responsible for about 68.6% of carbon emissions and taxis were responsible for 8.8%. A sufficiently efficient and simple emission calculation method will improve the efficiency of code operations when evaluating taxi environmental indicators. The VSP model can have more reliable accuracy when the taxi GPS sampling time is short and is widely used in micro-vehicle emissions.

#### 1.2.2. Urban Travel Carbon Emission Correlation Factors

Studies on the spatial heterogeneity of travel emissions and influencing factors tend to focus on traffic cells and grid areas. This is because the variability of relevant factors within different regions can be better reflected. Li et al. [19] studied the relationship between road density and emissions within TAZs by dividing Beijing city into 33 traffic analysis zones (TAZs) and found that emissions increased significantly when road density was greater than 0.6. Sun et al. [20] studied the association between population density (PD) and car usage (AU) and travel carbon emissions in the Ningbo region from a grid area perspective. The results showed significant spatial differences in the effects of PD and AU on travel emissions.

#### 1.2.3. Factors Related to the Efficiency of Taxi Operations

Taxi drivers’ experience judgment of active road sections at different periods is the key to continuous operation. Pick-up area, hotspot location and service location preferences had a positive impact on taxi drivers’ continuous pick-up and drop-off [21]. Zhang et al. [22] constructed a taxi service strategy matrix from the three perspectives of passenger search strategy, passenger delivery strategy and service area preference. Correlation between service strategy and revenue was evaluated and driver revenue was predicted through a policy matrix. Driven by the development of online car-hailing, some scholars hope to maximize the benefits of taxi drivers by balancing and redistributing supply and demand in different regions. Among them, the average delivery speed, cruising time, bus and subway operation were identified as important factors affecting revenue [23]. More factors are considered to enable taxi drivers to execute more efficient operational strategies.

#### 1.2.4. Method of Identifying Hotspots

To improve the efficiency of taxi operations, it is necessary to detect urban hotspot riding areas and predict their locations. Travel OD data have been used extensively in related studies. The attractiveness of the region to taxi drivers can be reflected in the cluster analysis of the OD data. Tang et al. [24] utilized the K-means clustering method to cluster OD data, and the travel distance, time and cost between different clustered areas were exploited to predict the spatial distribution of travel OD. The K-means clustering method cannot exclude the interference of noisy trajectory points. Therefore, more clustering methods are used to monitor hotspot areas. Zhou et al. [25] proposed a novel niche genetic algorithm (NGA) with density and noise for K-means clustering (NoiseClust). The results showed that NoiseClust had high performance and effectiveness in mining hotspots. Pan et al. [26] adopted the DBSCAN clustering algorithm to analyze the distribution of OD data on a finer scale and used it to classify land-use features. Shen et al. [27] utilized a similar clustering method to find hotspots for passengers getting on and off and used a trajectory clustering algorithm to evaluate the optimal route. Because the hotspot grid is determined in a small range, the k-means method, which reflects all the data information as much as possible, is selected.

### 1.3. Contributions

In summary, in the study of taxi fuel consumption and emissions, the focus is on the spatiotemporal distribution of emissions and their influencing factors. The data itself is rarely considered to provide a rationale for reducing emissions and optimizing the allocation of emissions space. These include limits on regional emissions or penalties for high-emission behavior in driving. Travel OD data are a convincing basis for the distribution of spatial and temporal taxi travel patterns. However, it is still hoped that more indicators can be exploited to improve driver efficiency. This includes different road factors, such as road morphology and intersection delays; state factors, such as speed and road congestion; and operational factors, such as average delivery speed and cruising period. However, environmental benefits are a part that is easily overlooked. When recommending hotspot areas for taxi drivers, it is often overlooked that areas with high travel demand bring more congested road sections, more competition for orders and denser emissions, which are unacceptable for drivers and transportation planners. We hope to meet the environmental benefits as much as possible while ensuring revenue when recommending road sections for taxi drivers.

This study exploited a gridding method to process the trajectory data and calculate the driver’s benefits and the emission indicators in the region. Spatial autocorrelation tests were applied to verify the variability of indicators for similar regions of travel demand. Then, the weights of these indicators were measured, and the hotspot grid was determined. Finally, the hotspot grid was clustered and analyzed to narrow the region to road segments. Figure 1 illustrates this process.

## 2. Data and Methods

### 2.1. Data

This study utilized taxi trajectory data from more than 4500 taxis in Nanchang for one day (9 July 2021). The taxi trajectory data included approximately 23.5 million records, containing longitude and latitude information for the taxi VehicleNum, State (vacant or occupied), Stime and Speed. A sample record is listed in Table 1. Moreover, the taxi trajectory data were recorded every 15 s with a positional accuracy of less than 10 m, which was acceptable for spatial analysis research.

Considering the accuracy of the calculation of indicators in the grid and the efficiency of code operation, GPS data needed to be cleaned. The data preprocessing method included five filtering rules. Clean data with the TransBigData Python package, version 0.4.12, developed by Dr. Yu Qing from Tongji University, Shanghai, China, about 10 million pieces of data were retained. A sample of the processed data is given in Table 2. 

Remove data whose field value is null.Remove data with duplicate field values.Remove data beyond the scope of Nanchang City.Remove data at speeds over 80 km/h.Remove State conversion error data. For example, the OpenStatus value is 0 continuously, except for one trajectory point, which is 1.

The original taxi trip records were obtained by analyzing the transformation of the status field of the trajectory data. The State field was converted from 0 to 1 and then to 0 was recorded as a trip record. These records constituted the OD table of Nanchang taxi trips in a day, of which about 90,000 records were generated. The OD records of taxi trips are given in Table 3. The data for trips that were too short (order time less than 1 min) were removed, which was difficult in real situations.

### 2.2. Methods

#### 2.2.1. Carbon Emission Accounting by Taxi Travel

Vehicle specific power (VSP) models have been commonly used in eco-driving-related studies to estimate vehicle output power. Early VSP models used vehicle speed, acceleration and road type to estimate fuel consumption. It is challenging to obtain the operating state of vehicles due to the high data requirements [16]. Scholars have experimentally determined the relationship between average vehicle speed and fuel consumption, engine and emissions to improve the VSP model [28]. This method has been used by scholars to calculate the spatial and temporal characteristics of emissions generated by taxi trips in urban areas [20]. The fuel consumption was calculated as follows:(1)$$F_{i,j}=\sum_{l=1}^{n}ER_{0}\cdot NFCR_{l}\cdot T_{i,l}$$
where $F_{i,j}$ represents the total fuel consumption of taxi i from the origin to the destination; n represents the number of all records from the origin to the destination; $ER_{0}$ represents the average fuel consumption rate of the taxi, which is set as 0.274 (g/s) [29]; $NFCR_{l}$ represents the normalized fuel consumption rate in the average speed interval l, The value of the normalized fuel consumption rate corresponding to the vehicle speed range is obtained by referring to the relevant study [16,20]; $T_{i,l}$ represents the driving duration (s) in the intermediate speed interval l of taxi i.

Since there is no significant difference between vehicle type and engine size of taxis in the study area, the same emission factors were exploited to calculate the emissions generated by taxi trips. The total carbon emissions of taxi trips were obtained according to Equation (1).
(2)$$C_{i,j}=EF_{\alpha }\cdot F_{i,j}=EF_{\alpha }\sum_{l=1}^{n}ER_{0}\cdot NFCR_{l}\cdot T_{i,l}$$
where $C_{i,j}$ (kg) represents the trip’s carbon emissions by taxi i, and $EF_{\alpha }$ is the emission factor of exhaust α, which is 2.18 kg/L [12,16,20].

#### 2.2.2. Calculation of Indicators in the Grid Area

The common schemes for analyzing taxi carrying behavior in the grid area are the square grid-based method and the Tyson polygon method. A square grid of 1000 × 1000 m was utilized in this study. The calculation formula is as follows:(1)Calculate the increase in latitude and longitude of each grid:
(3)$$\Delta Lon=\frac{x\cdot 360}{2\pi \cdot R_{e}\cdot \cos\left(\frac{\left(lat_{1}-l_{1}t_{2}\right)\cdot \pi }{360}\right)}$$
(4)$$\Delta Lat=\frac{x\cdot 360}{2\pi \cdot R_{e}}$$

(2)Calculate the coordinate ID of each grid:

(5)$$LatID=\frac{m-\left({\;\mathrm{lon}}_{1}-\frac{\Delta Lon}{2}\right)}{\Delta Lon}$$(6)$$LatID=\frac{n-\left({\;\mathrm{lat}}_{1}-\frac{\Delta Lat}{2}\right)}{\Delta Lat}$$where $R_{e}$ represents radius of the earth, *m* represents the longitude of the point that inputted, n represents the latitude of the point that was inputted.

**Definition** **1.***The grid traffic state index (G-tsi) is defined as a comprehensive index for evaluating the traffic congestion state. This is the relative ratio of free-flowing traffic speed and actual traffic speed over some time and can be expressed by the formula*.

(7)$$G\_tsi=\frac{V_{f}-V_{t}}{V_{f}}=1-\frac{V_{t}}{V_{f}}$$where $V_{t}$ denotes the average speed of the taxi in the studied time grid, $V_{f}$ represents the free-flow speed, and the average vehicle speed of the whole day is used as the free-flow speed in this study.

**Definition** **2.***Grid boarding points (G-bp) are defined as the level of trip activity within each study grid. This is the cumulative number of trip origins over some time and can be expressed by the formula*.

(8)$$G\_bp=\sum_{\left(x_{i},y_{i}\right)\in \left\{x_{n},y_{m}\right\}}M_{od}$$where $\left(x_{i},y_{i}\right)$ is the trajectory point in grid number $\left\{x_{n},y_{m}\right\}$, and $M_{od}$ is the starting point of the trip.

**Definition** **3.**
*Grid order acquisition probability (G-oap) is defined as the probability of a taxi acquiring an order in each study grid. This is the ratio of the travel origin to the number of idle track points over time. This can be expressed in the following formula:*


(9)$$G\_\mathrm{oap}=\frac{\sum_{\left(x_{i},y_{i}\right)\in \left\{x_{n},y_{m}\right\}}M_{od}}{\sum_{\left(x_{i},y_{i}\right)\in \left\{x_{n},y_{m}\right\}}M_{null}}$$where $M_{null}$ represents trajectory points with no passengers.

**Definition** **4.***Grid order revenue (G-or) is defined as the average revenue per minute of orders within each study grid, reflecting the average level of order revenue obtained from that grid. It is the ratio of the total value of orders generated in the grid over some time to the product of the number of orders and order time and can be expressed by the formula*.

(10)$$G\_\mathrm{oap}=\frac{\sum_{\left(x_{i},y_{i}\right)\in \left\{x_{n},y_{m}\right\}}P_{od}}{N*T_{Avg}}$$where $P_{od}$ represents order revenue, calculated by regional tariff and operating mileage; N represents the number of orders and $T_{Avg}$ represents the average duration of orders originating from the study grid.

**Definition** **5.**
*Grid carbon emissions (G-c) and grid average carbon emissions (G-avgc) were used to reflect regional emissions. The former is influenced by the number of trips and the latter is influenced more by the average vehicle speed in the region. The total emissions are calculated using the adjacent trajectory points retained in the grid and divided by the total number of trajectory points in the grid to calculate the average emissions for each adjacent trajectory point. The calculation formula is as follows:*


(11)$$G\_c=\sum_{\left(x_{i},y_{i}\right)\in \left\{x_{n},y_{m}\right\}}c_{i,j}$$(12)$$G\_avgc=\frac{\sum_{\left(x_{i},y_{i}\right)\in \left\{x_{n},y_{m}\right\}}c_{i,j}}{\sum_{\left(x_{i},y_{i}\right)\in \left\{x_{n},y_{m}\right\}}Mp}$$where $c_{i,j}$ represents adjacent trajectory points travel emissions, calculated from Equation (2), Mp represents adjacent trajectory points.

#### 2.2.3. Comprehensive Evaluation Methodology

The entropy weight method (EWM) calculates an objective weight for each indicator according to the discrete degree of the data [30]. The Technique for Order Preference by Similarity to an Ideal Solution (TOPSIS) method ranks a limited number of evaluation objects according to their proximity to an idealized target [31]. In this study, the entropy weight method is used to calculate the weights of various indicators in the grid, and the comprehensive evaluation value is calculated by TOPSIS. Thus, grids with higher overall scores in each period are found.

#### 2.2.4. K-Means Clustering Method

Clustering is a common unsupervised learning method in which similar data samples are grouped into clusters. The clustering of taxi pick-up points in the hotspot grid can identify hotspot road sections. The K-means algorithm is a classical clustering algorithm in machine learning technology that relies on the distance between the pick-up points as the evaluation index in the process of clustering. The closer the distance, the higher the similarity. Combine these close pick-up points to form a cluster and take these composed clusters as the final target. Geoda software was used to perform cluster analysis and visualize the processed data. The algorithm flow is shown in Figure 2.

## 3. Results

Nanchang, located in central China, is the capital city of Jiangxi Province. Nanchang’s per capita GDP was 143,157 RMB (about 16,083 USD) in 2021 and is one of the first provincial capitals in the central region to be piloted as a low-carbon city in China. The taxi business in Nanchang is responsible for the transportation needs of the core areas of Nanchang. The monthly income of taxi drivers is mostly 6000–8000 RMB, which is lower than the per capita income level in Nanchang. According to the Nanchang City Development and Reform Commission published by the Nanchang City taxi tariff. The starting price is 8 RMB/km, the per kilometer price is 2.1 RMB/km, the driving distance exceeds 8 km though the part of 50% of the emptying subsidy, night 11:00 p.m. to 5:00 a.m. plus 20% of the kilometer subsidy fee. The order revenue is calculated by this in this study. The study area is located in the administrative area of Nanchang city (light green in Figure 3, total 7553 grids). The main passenger-carrying areas of taxis are located in Xihu District, Qingyunpu District and Qingshan Lake District (light red in Figure 3, 196 grids), and the grid unit is 1000 × 1000 m.

### 3.1. Taxi Statistics Indicators in Each Period

Travel demand by taxi travel varied greatly during the day. Figure 4 shows that the travel activity peaked at 9:00 a.m. to 11:00 a.m., and the lowest value appeared between 4:00 a.m. and 6:00 a.m. Travel activity was relatively inactive between 12:00 p.m. and 2:00 p.m. Most Chinese people were used to rest during this period. The number of taxis was consistent with the trend of GPS data volume. Figure 5 shows that the overall order duration was maintained at 5–20 min, which is in line with the actual situation. Orders from 7:00 a.m. to 9:00 a.m. and 5:00 p.m. to 7:00 p.m. took longer, about 5 min, which may be due to more congested roads. To study the temporal characteristics of taxi trips, four periods were selected. The length of the extracted period was set to two hours concerning the duration of the peak hours. Morning and evening peak periods (7:00 a.m. to 9:00 a.m., 5:00 p.m. to 7:00 p.m.), early morning period (1:00 a.m. to 3:00 a.m.), and night period (8:00 p.m. to 10:00 p.m.). 

The reasonableness of the number and operation of taxis is reflected by the taxi idling rate, which is calculated by calculating the ratio of the idling points to passenger carrying points. Figure 6 shows that the idle rate was lowest in the morning and evening peak periods (1:00 a.m. to 3:00 a.m. and 5:00 p.m. to 7:00 p.m.), but the idle rate was over 50% throughout the day and even close to 80% in the early morning period (1:00 a.m. to 3:00 a.m.). Taxi drivers were looking for passengers more than half of the time. The order revenue by taxi travel varied during the day (Figure 7). The morning and evening peak periods were the easiest time to get orders but also the least efficient time for revenue. The average order revenue was about 1.1 RMB/min, during daytime off-peak periods, this value was about 1.4–1.5 RMB/min, in the early morning period exceeding 1.8 RMB/min. Compared with the peak periods, about 30–60% were higher. In addition, the average distance traveled throughout the day was about 7.5 km, and the income per trip during the daytime period was about 22 RMB. The early morning periods are higher than daytime periods, up to 31 RMB. 

Carbon Emission Per Kilometers (CEPK) is a global standard metric that provides a more objective assessment of carbon emissions and is calculated by dividing total travel emissions by total travel distance. Figure 8 shows that the CEPK remained around 0.25 kg/km despite the large variation in total carbon emissions per hour. During the 4:00 a.m. to 5:00 a.m. period, the total emissions were 4570 kg, which increased to 16,813 kg during the peak period, an almost 4-fold increase. This figure decreased slightly from 12:00 p.m. to 2:00 p.m. noon and increased again in the afternoon, with a total emission of about 300 tons throughout the day. Combined with Figure 6, it can be found that a large part of Nanchang taxi emissions come from idle trips.

### 3.2. Taxi Statistics Indicators in Each Grid

The three aspects of traffic condition, driver revenue, and carbon emission in the grid were subdivided into G-bp, G-tsi, G-oap, G-or, G-c, and G-avgc. The distribution characteristics of the indicators in the grid and their dynamics over time were explored. (periods selected were 1:00 a.m. to 3:00 a.m., 7:00 a.m. to 9:00 a.m., 5:00 p.m. to 7:00 p.m., and 8:00 p.m. to 10:00 p.m. in Figure 9, Figure 10, Figure 11, Figure 12 and Figure 13)

As shown in Figure 9. According to the latitude and longitude of the grid center, the main aggregation locations were distributed near Wanda Square in Honggu Tan District and Bayi Square in Donghu District. The number of high travel demand grids (>120) increased with time, and the highest travel demand grids existed in the period 8:00 p.m. to 10:00 p.m. Two grids with significantly deeper colors appeared in the periods of 5:00 p.m. to 7:00 p.m. and 8:00 p.m. to 10:00 p.m., which were located in the area where Nanchang Changbei International Airport and Nanchang West Station. The G-tsi had obvious period variation characteristics (Figure 10). The urban traffic congestion was more serious in the morning and evening peak periods, with about 880 grids with G-tsi greater than 0 and 123 grids exceeding 0.25. The early morning period had better traffic conditions, with 445 grids with G-tsi greater than 0 and 114 grids exceeding 0.25. The night periods are at the middle level. The traffic congestion grids scattered in suburban areas had too little data, and at some periods, there were no taxis passing through. Therefore, the statistical value of these areas was not high compared to the congested areas in the city center.

The G-oap had spatial heterogeneity in the grid distribution. An interesting conclusion from comparing Figure 9 was that high travel demand areas were not high-order acquisition probability areas. Figure 11 shows that the G-oap of the grids with higher travel demand were almost all in the 50% to 90% range of the overall data, but the distribution of the easiest areas to acquire orders (90% to 99%) appeared scattered. The early morning period was the most difficult time to obtain orders and the number of grids was also the least. Relatively low revenue in the central city and higher revenue in the suburbs (Figure 12). Revenue from travel hotspots was generally distributed between 10% and 50% of the overall data. This means that, although these areas generated more orders, their earnings during this period were not considerable compared to other regions. Taking orders at night had the highest revenue because extra night surcharges were charged at night. Morning and evening peak periods were the lowest revenue periods for taxi drivers. Even though there was already a high range of travel demand at this time, poor traffic conditions still had a negative impact on drivers’ revenue efficiency. This disturbance was most pronounced during the evening peak period.

The grids with CO_2_ emissions data were divided into four groups by value from high (group 1) to low (group 4). Table 4 lists the specific values of each group’s emissions and the proportion of total emissions. Figure 13 shows the distribution of grid emissions at different periods. There were differences in different periods and groups, but the emissions were still concentrated in a few hotspot areas. The carbon emissions of taxi travel presented a step distribution phenomenon in space, decreasing from the high-emission core area to the suburbs. Airports and train stations are isolated, high travel emission areas. Group 1 accounted for more than one-third of the total emissions and even approached half of the total emissions during the night. High-emission grids contributed 57% to 70% of travel emissions, with about 2% to 4% of the area. Low-emission areas generated 5% to 8% of travel emissions, with about 70% to 80% of the area. The average grid carbon emissions for the taxi trips showed opposite distribution characteristics (Figure 14). Most of the areas with high average emissions were suburbs, which is consistent with previous studies [20]. The lower emission levels between trajectory points in the lower vehicle speed area also mean that more travel time was generated in the grid. Among the emission indicators, we chose G-c as a comprehensive evaluation indicator because it was more suitable for describing regional travel emissions. G-avgc had a small gap in the high travel demand grids.

### 3.3. Spatial Autocorrelation Analysis

To visualize the contrast of the results more clearly, the study adopted the method of graphing two types of interval data (numerical and percentage) to enhance the color contrast. However, the aggregation phenomenon was not supported by theory. Moran’s I is an indicator of the spatial correlation of the data [12]. It was used to evaluate the spatial distribution characteristics of the indicators within the grid. Table 5 lists the test results of the previous five indicators. The Z-score is greater than 2.58 and the P-Value is less than 0.01, indicating that the data are statistically significant, and the distribution of the indicators is not random. There was a significant positive spatial dependence between G-c and G-bp, indicating that areas with a higher travel demand were more likely to cluster. The Moran’s I value of G-bp was higher in the morning peak period and night period; residents had a higher demand for travel. G-tsi also had a positive spatial correlation, but the Moran’s I value was small. It had a certain spatial aggregation phenomenon, but this characteristic was not significant. The G-oap full-time test did not hold, indicating that there was no spatial aggregation phenomenon. The spatial distribution of this indicator in the grid was random. The results of the G-or test showed time-period differences. In the taxi industry, areas with high travel demands were significantly clustered, but there may be differences in other indicators in high-demand grids. Therefore, we can determine the comprehensive optimal grid area for factors such as travel demand, revenue and emissions in a certain period through a comprehensive comparison.

### 3.4. Comprehensive Evaluation of Grid Indicators

The study exploited G-bp, G-oap, G-or, G-tsi, and G-c, five indicators to determine the recommended grids (G-tsi and G-c are negative indicators). Due to the high concentration of taxi activities in Nanchang, demand varies greatly. The travel demand varies greatly between grids, and the top two grids (17 to 35 grids) with the largest travel volume in Figure 9 were selected and comprehensively evaluated. As shown in Figure 15. The weighting of the indicators varied from period to period, but (G-bp) travel demand remained the most important factor. The lowest weight is 0.33 in the period 1:00 a.m. to 3:00 a.m., and the highest is 0.48 in the period 7:00 a.m. to 9:00 a.m. The secondary weight is different in each period. In the period 5:00 p.m. to 7:00 p.m., G-or became the secondary weight (0.28), in the period 7:00 a.m. to 9:00 a.m. G-c became the secondary weight (0.21), and in the period 8:00 p.m. to 10:00 p.m. G-oap became a secondary weight (0.27). The comprehensive ranking of the grid was determined by the weight of the multiple indicators. For example, the grid numbered (68,84) in the period from 5:00 p.m. to 7:00 p.m. was not in high-trip demand. However, the relatively remote location allowed drivers to earn more emptying subsidies and higher travel speeds, resulting in higher rankings in this area than in other areas during the same period. Table 6 lists the top five grid areas and the specific values of their indicators in the four periods. The high-ranking grids in different periods were not fixed, but the spatial positions of the recommended grids were relatively close. 

### 3.5. Recommended Road Sections

Some higher-ranked grids were selected in each period and cluster analysis was performed on the travel origins. Table 7 lists the clustering results for some grids. These sections are worth being recommended for idling taxis. The starting point of the trip was clustered around the stations, hotels, hospitals and shopping malls. According to the recommendations in this study. In the evening peak period, the nearest idle drivers can consider the section of Nanchang West Station Shizhong Mountain with grid numbers (56,58). In the early morning period, look for passengers in the section near People’s Park in grid number (68,65), and in the night period, look for passengers in consumer places such as Parkside Shopping Center and Wanda Plaza in grid number (66,64). 

## 4. Discussion

Our results show that taxi travel hotspots that combine emission and revenue factors are not randomly assigned at different periods. The proposed method provides new insights into the environmental and benefit optimization of taxis.

The gridded spatiotemporal pattern of taxi trips in a day was observed based on the number of taxis and the order consumption time. The results confirmed that there was a serious phenomenon of idle taxis in the region, which was quite unfavorable to the construction of a low-carbon city. The actual operating income of the drivers was calculated, confirming that there were spatial and temporal differences. Higher CEPK values and total emissions require policy planners to enact stricter carbon reduction regulations and policies. These analyses confirm the need for a rational approach to reconciling the supply–demand balance between drivers and passengers, as well as excessive emissions in some areas.

Visual analysis of the indicators in the grid area. The distribution of different indicators in the area was clearly observed, and travel demand was highly concentrated in the core urban area. Such agglomeration results in a considerable concentration of vehicle emissions that are harmful to the health of residents living in these areas. We found that the G-oap distribution is quite random, and high travel demand areas did not mean easier access to orders. Active areas close to outer urban areas tended to provide more order revenue, such as Changbei International Airport. Vehicles in the core urban area were more congested, but there were still differences in the G-tsi of the grid in the core area. By comparing the spatial and temporal distribution characteristics of total emissions and average emissions in the analysis of taxi emissions, it was found that the determinant of regional emissions was the number of trips in the region. It will take an aggressive shift in travel patterns to achieve real reductions in emissions.

Travel demand and emissions were spatially autocorrelated, and both occurred in the travel network as an aggregation phenomenon. G-or and G-oap were more randomly distributed to some extent. Through the weight calculation of the indicators, it was found that travel demand was still the most important factor in evaluating a grid. The rankings of the secondary indicators were dynamic changes. The recommended hotspots are not fixed. In the evening period, entertainment and consumption places become the main recommended area, and for the evening peak period, it is more recommended to go to bus stations, railway stations and other transportation hubs. The research provides the recommended road segments after clustering in different periods.

Urban taxis have gradually shifted to taking orders by mobile phone in China. This effectively improves the efficiency of passenger travel demand matching. However, it also requires drivers to know where and when there may be a high trip demand to minimize the phenomenon of idle driving. In the actual operation of taxis in Nanchang, there was still a high idle driving situation, which confirmed that drivers still lacked judgment on hotspots. This is a key factor that affects the driver’s income. For this reason, this study hopes to exploit this method to provide recommendations for the road section where the driver is idle and the area is searching for passengers. To increase the emission indicator is to hope that the government can formulate corresponding carbon subsidy policies or restrictive policies. Although the electrification and zero-carbonization of transportation is considered the most reliable way to reduce transportation-related carbon emissions, this process will still take a long time. Therefore, the available resources should be fully utilized to adjust the level of urban travel emissions. The subsidy or limit is designed to reward drivers who meet the rules to reduce their existing emissions levels. In addition, indicators available for this hotspot section can be used as the basis for adjusting the distribution of travel modes in the area, balancing existing travel modes, such as bus and shared bicycle stations in the area. Reasonable adjustment of travel patterns in the high-emission grid area will help to change the unreasonable concentration of emissions in the area. 

The current study has some problems but also has the potential to continue further study. Whether the study of emissions and income at fine spatial scales affects the commonality of these indicators on a larger scale cannot be confirmed at this time. The travel chain formed by OD data are not considered; it only provides time-based road segment recommendations. 

## 5. Conclusions

We study the distribution of hotspots for taxi trips at a finer scale combining carbon emissions and benefits. The proposed method is based on an actual taxi trip case in Nanchang, China. Six indicators related to revenue and carbon emissions are defined and measured to reveal the spatial and temporal characteristics of taxi emissions and revenue. Comprehensive evaluation and clustering methods confirm that there are constant hotspots in taxi operations, which can effectively replace the empirical judgment of drivers looking for passengers. In addition, the configuration optimization of low-carbon or zero-carbon transportation in areas with high emission phenomenon in non-constant hotspots is also worthy of attention of traffic managers. In future studies, road network and point of interest (POI) information will be explored as influencing factors in relation to grid indicators and will define the grid scale. All-day OD data of drivers with low emissions and high incomes will be used to determine complete route recommendations through trajectory clustering. This will enhance the applicability of such methods in more cities.

## Figures and Tables

**Figure 1 ijerph-19-11490-f001:**
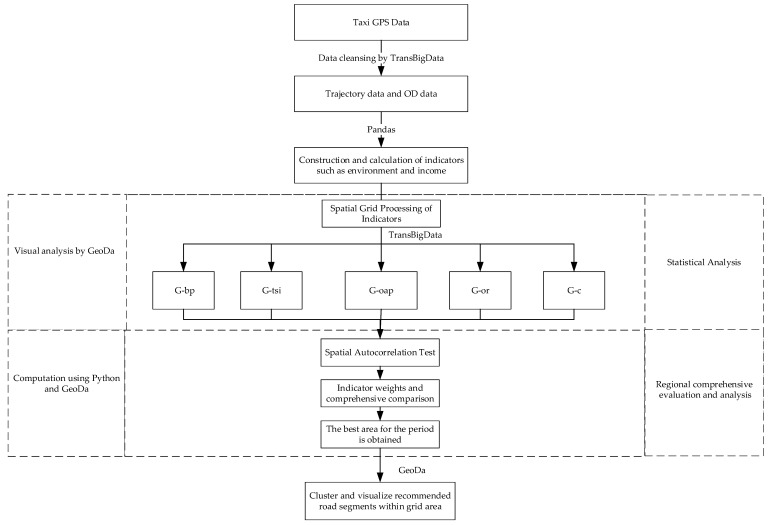
Research Framework.

**Figure 2 ijerph-19-11490-f002:**
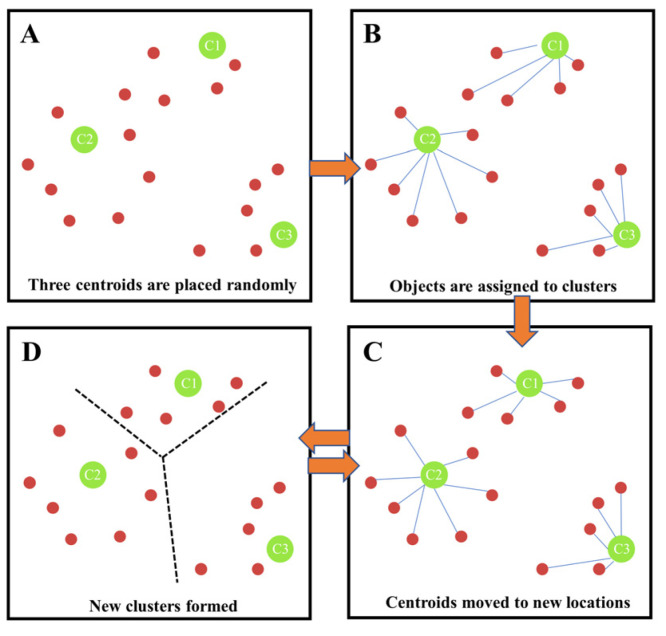
k-means clustering algorithm flow.

**Figure 3 ijerph-19-11490-f003:**
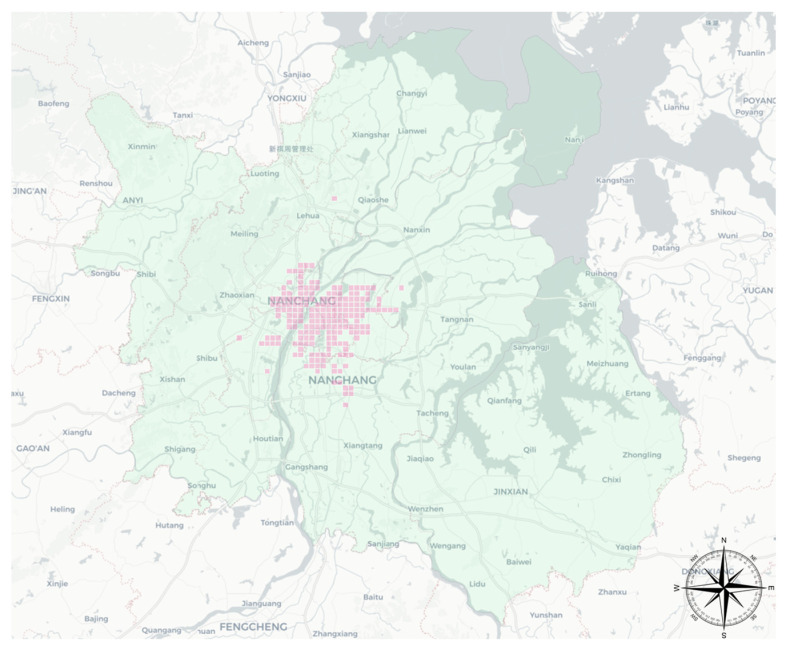
Study area in Nanchang.

**Figure 4 ijerph-19-11490-f004:**
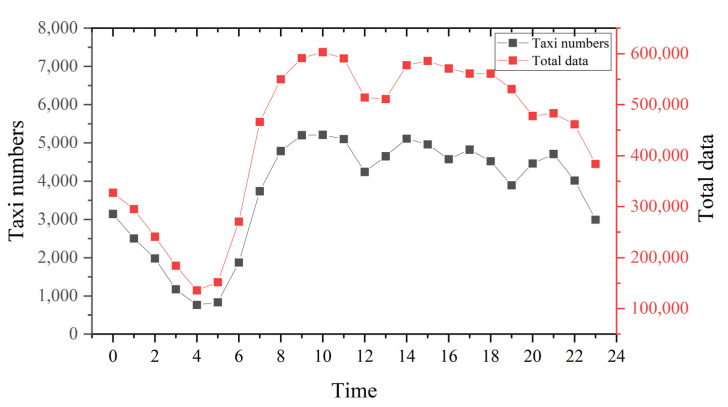
Number of taxis and data volume.

**Figure 5 ijerph-19-11490-f005:**
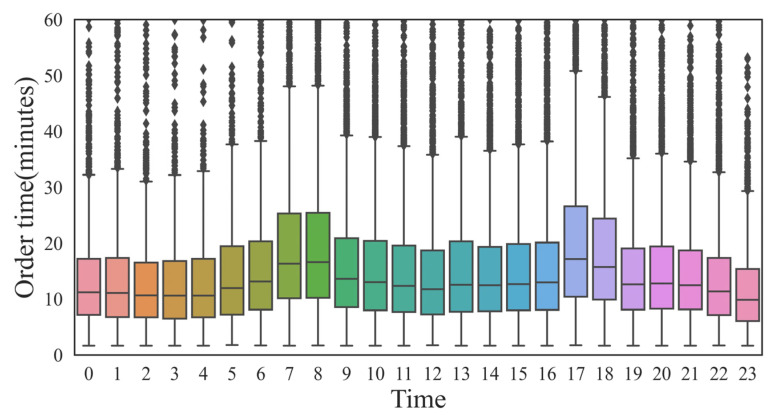
Order time consumption.

**Figure 6 ijerph-19-11490-f006:**
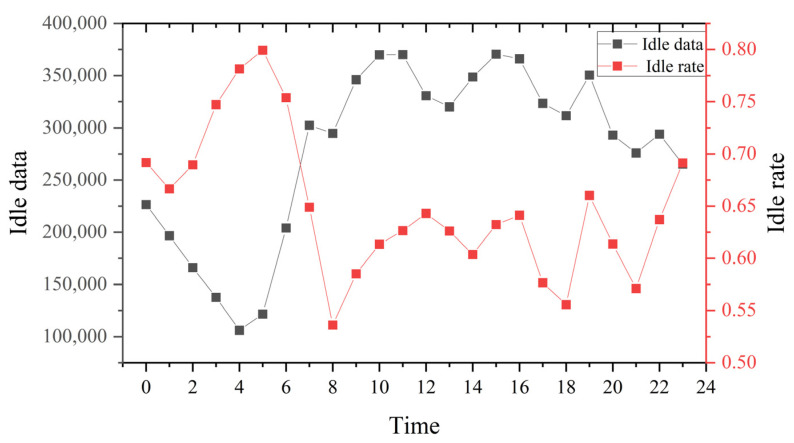
Taxis idling data volume and idling rate.

**Figure 7 ijerph-19-11490-f007:**
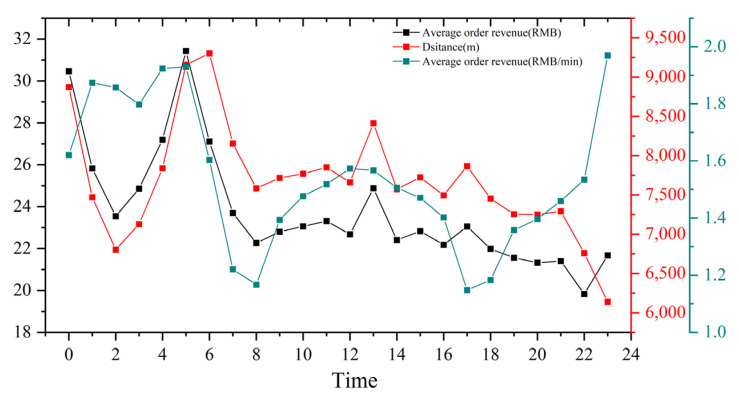
Order revenue and distance.

**Figure 8 ijerph-19-11490-f008:**
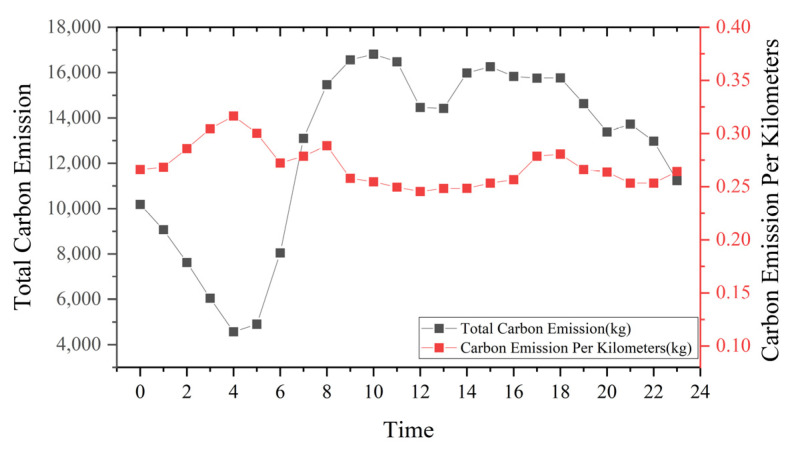
Carbon emissions and average per kilometer.

**Figure 9 ijerph-19-11490-f009:**
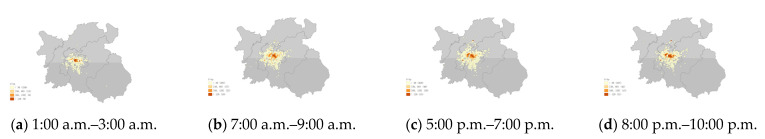
(**a**) Grid boarding points (G-bp) of each grid area at 1:00 a.m.–3:00 a.m. (**b**) Grid boarding points (G-bp) of each grid area at 7:00 a.m.–9:00 a.m. (**c**) Grid boarding points (G-bp) of each grid area at 5:00 p.m.–7:00 p.m. (**d**) Grid boarding points (G-bp) of each grid area at 8:00 p.m.–10:00 p.m.

**Figure 10 ijerph-19-11490-f010:**
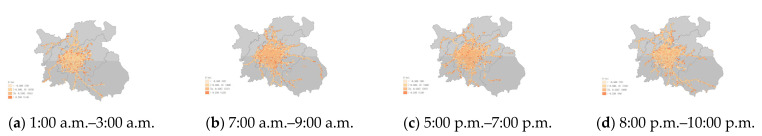
(**a**) Grid traffic state index (G-tsi) of each grid area at 1:00 a.m.–3:00 a.m. (**b**) Grid traffic state index (G-tsi) of each grid area at 7:00 a.m.–9:00 a.m. (**c**) Grid traffic state index (G-tsi) of each grid area at 5:00 p.m.–7:00 p.m. (**d**) Grid traffic state index (G-tsi) of each grid area at 8:00 p.m.–10:00 p.m.

**Figure 11 ijerph-19-11490-f011:**
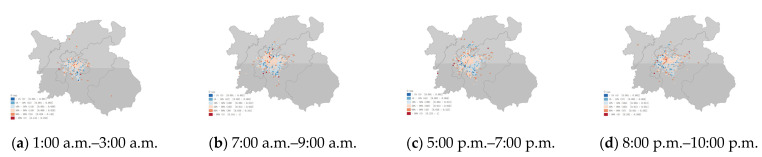
(**a**) Grid order acquisition probability (G-oap) of each grid area at 1:00 a.m.–3:00 a.m. (**b**) Grid order acquisition probability (G-oap) of each grid area at 7:00 a.m.–9:00 a.m. (**c**) Grid order acquisition probability (G-oap) of each grid area at 5:00 p.m.–7:00 p.m. (**d**) Grid order acquisition probability (G-oap) of each grid area at 8:00 p.m.–10:00 p.m.

**Figure 12 ijerph-19-11490-f012:**
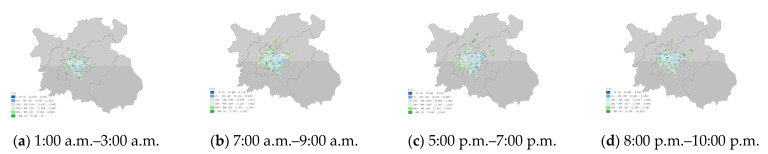
(**a**) Grid order revenue (G-or) of each grid area at 1:00 a.m.–3:00 a.m. (**b**) Grid order revenue (G-or) of each grid area at 7:00 a.m.–9:00 a.m. (**c**) Grid order revenue (G-or) of each grid area at 5:00 p.m.–7:00 p.m. (**d**) Grid order revenue (G-or) of each grid area at 8:00 p.m.–10:00 p.m.

**Figure 13 ijerph-19-11490-f013:**
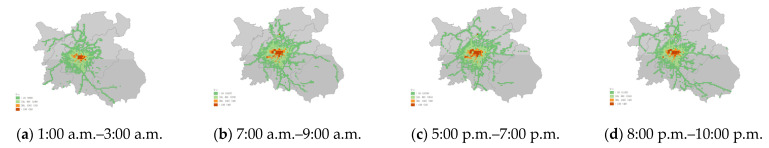
(**a**) Grid carbon emissions (G-c) of each grid area at 1:00 a.m.–3:00 a.m. (**b**) Grid carbon emissions (G-c) of each grid area at 7:00 a.m.–9:00 a.m. (**c**) Grid carbon emissions (G-c) of each grid area at 5:00 p.m.–7:00 p.m. (**d**) Grid carbon emissions (G-c) of each grid area at 8:00 p.m.–10:00 p.m.

**Figure 14 ijerph-19-11490-f014:**
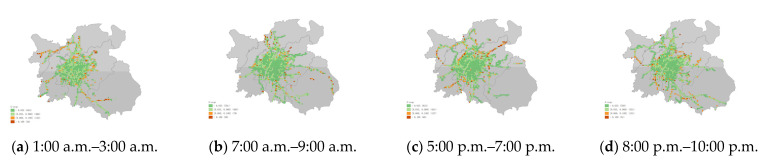
(**a**) Grid average carbon emissions (G-avgc) of each grid area at 1:00 a.m.–3:00 a.m. (**b**) Grid average carbon emissions (G-avgc) of each grid area at 7:00 a.m.–9:00 a.m. (**c**) Grid average carbon emissions (G-avgc) of each grid area at 5:00 p.m.–7:00 p.m. (**d**) Grid average carbon emissions (G-avgc) of each grid area at 8:00 p.m.–10:00 p.m.

**Figure 15 ijerph-19-11490-f015:**
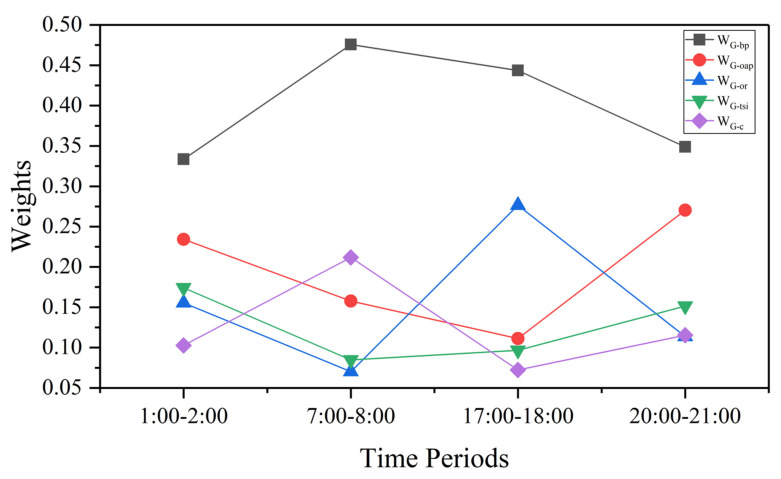
Time weights of each indicator.

**Table 1 ijerph-19-11490-t001:** Sample trajectory data.

	VehicleNum	Stime	Longitude	Latitude	Angle	Speed	State
0	85939	7/9/2021 23:07:50	115.885883	28.685583	164	34.6	0
1	122588	7/9/2021 23:07:50	115.924966	28.600300	174	0	0
…	…	…	…	…	…	…	…

**Table 2 ijerph-19-11490-t002:** Sample of trajectory data after data cleaning.

	VehicleNum	Stime	Longitude	Latitude	Speed	State
0	66652	7/9/2021 00:00:37	115.935933	28.669550	39.7	0
1	66652	7/9/2021 00:01:07	115.935966	28.669550	0	0
…	…	…	…	…	…	…

**Table 3 ijerph-19-11490-t003:** Taxi trip record data.

	VehicleNum	Stime	Slongitude	Slatitude	Etime	Elongitude	Elatitude
0	66652	2021/7/9 0:33	115.935966	28.669550	2021/7/9 0:42	115.944200	28.651283
1	66652	2021/7/9 7:46	115.935650	28.655167	2021/7/9 7:52	115.920066	28.663583
	…	…	…	…	…	…	…

**Table 4 ijerph-19-11490-t004:** Share of emissions and total emissions for each group of grids.

Time	Group 1 (Highest)	Group 2	Group 3	Group 4 (Lowest)	Total (kg)
1:00 a.m.–3:00 a.m.	6313.4979 (37.82%)	3500.3536 (20.97%)	5599.5307 (33.55%)	1278.2973 (7.66%)	16691.6797
7:00 a.m.–9:00 a.m.	12,101.1215 (42.36%)	6025.9213 (21.09%)	8944.7353 (32.31%)	1497.1840 (5.24%)	28,568.9621
5:00 p.m.–7:00 p.m.	15,300.6980 (48.54%)	6213.9535 (19.71%)	8435.8982 (26.76%)	1574.3726 (4.99%)	31,524.9224
8:00 p.m.–10:00 p.m.	13,446.5223 (49.60%)	4357.9346 (16.08%)	7747.6962 (28.58%)	1557.0271 (5.74%)	27,109.1804

**Table 5 ijerph-19-11490-t005:** Spatial autocorrelation analysis of indicators within each period.

Indicators	Time	Moran’I	Z-Score	*p*-Value
G-bp	1:00 a.m.–3:00 a.m.	0.4270	11.6336	*p* ≤ 0.01
	7:00 a.m.–9:00 a.m.	0.6159	21.1729	*p* ≤ 0.01
	5:00 p.m.–7:00 p.m.	0.4226	15.4093	*p* ≤ 0.01
	8:00 p.m.–10:00 p.m.	0.5248	15.3427	*p* ≤ 0.01
G-tsi	1:00 a.m.–3:00 a.m.	0.0796	4.24850	*p* ≤ 0.01
	7:00 a.m.–9:00 a.m.	0.1111	6.45240	*p* ≤ 0.01
	5:00 p.m.–7:00 p.m.	0.0674	3.78570	*p* ≤ 0.01
	8:00 p.m.–10:00 p.m.	0.1416	8.22490	*p* ≤ 0.01
G-oap	1:00 a.m.–3:00 a.m.	0.1134	2.97010	*p* ≥ 0.01
	7:00 a.m.–9:00 a.m.	0.0382	1.70660	*p* ≥ 0.01
	5:00 p.m.–7:00 p.m.	0.0105	0.68050	*p* ≥ 0.01
	8:00 p.m.–10:00 p.m.	0.0070	0.29770	*p* ≥ 0.01
G-or	1:00 a.m.–3:00 a.m.	0.0149	0.45000	*p* ≥ 0.01
	7:00 a.m.–9:00 a.m.	0.3211	10.5107	*p* ≤ 0.01
	5:00 p.m.–7:00 p.m.	0.3705	12.8477	*p* ≤ 0.01
	8:00 p.m.–10:00 p.m.	0.1020	4.93820	*p* ≤ 0.01
G-c	1:00 a.m.–3:00 a.m.	0.7421	37.5382	*p* ≤ 0.01
	7:00 a.m.–9:00 a.m.	0.7218	41.9020	*p* ≤ 0.01
	5:00 p.m.–7:00 p.m.	0.6741	39.1429	*p* ≤ 0.01
	8:00 p.m.–10:00 p.m.	0.7141	43.0642	*p* ≤ 0.01

**Table 6 ijerph-19-11490-t006:** Weighted grid value ranking for each time period(The bold part is the relevant indicators of the grid area that ranks first in the comprehensive hotspot during the period).

Time	Grid Nember	Grid Center	G-bp	G-oap	G-or	G-tsi	G-c	C_i_	Rank
1:00 a.m.–3:00 a.m.	**(66,65)**	**(115.894023,28.680062)**	**326**	**0.03692**	**1.8498**	**−0.1016**	**441.959**	**0.701**	**1**
	(68,65)	(115.914542,28.680062)	341	0.03651	1.5939	0.0215	393.908	0.620	2
	(66,64)	(115.894023,28.671068)	238	0.02450	1.7812	−0.2824	412.398	0.560	3
	(65,64)	(115.883763,28.671068)	187	0.03012	1.8924	−0.2047	290.990	0.548	4
	(65,63)	(115.883763,28.662075)	182	0.02282	2.0188	−0.1932	323.389	0.495	5
7:00 a.m.–9:00 a.m.	**(68,62)**	**(115.914542,28.653082)**	**271**	**0.02014**	**1.1706**	**0.2036**	**557.650**	**0.680**	**1**
	(68,63)	(115.914542,28.662075)	216	0.01851	1.1388	0.0469	344.384	0.626	2
	(65,65)	(115.883763,28.680062)	198	0.01997	1.0526	0.1908	383.916	0.538	3
	(67,63)	(115.904282,28.662075)	185	0.02176	1.2976	0.0692	379.707	0.538	4
	(66,65)	(115.894023,28.680062)	181	0.01941	1.0567	0.1875	485.110	0.480	5
5:00 p.m.–7:00 p.m.	**(56,58)**	**(115.791424,28.617109)**	**606**	**0.02751**	**0.9885**	**0.0663**	**715.828**	**0.620**	**1**
	(68,84)	(115.914542,28.850933)	157	0.01477	2.6009	−0.0285	330.467	0.449	2
	(67,65)	(115.904282,28.680062)	347	0.02784	1.0767	0.1379	674.375	0.421	3
	(67,64)	(115.904282,28.671068)	273	0.02488	1.0501	0.1202	570.395	0.332	4
	(66,64)	(115.894023,28.671068)	268	0.02817	0.9910	0.2121	467.906	0.321	5
8:00 p.m.–10:00 p.m.	**(62,66)**	**(115.852983,28.689055)**	**270**	**0.03747**	**1.3368**	**0.0544**	**302.467**	**0.667**	**1**
	(56,58)	(115.791424,28.617109)	333	0.02851	0.9230	0.0378	353.311	0.651	2
	(66,64)	(115.894023,28.671068)	359	0.02267	1.2701	0.0654	602.324	0.589	3
	(65,65)	(115.883763,28.680062)	251	0.02531	1.2667	−0.0025	434.365	0.532	4
	(63,66)	(115.863243,28.689055)	179	0.03161	1.3689	0.0030	276.819	0.508	5

**Table 7 ijerph-19-11490-t007:** Highly ranked grid passenger boarding point clustering centers and recommended roadways (The bolded part is the information about the hotspot section that ranks first in the period).

Time	Grid Number	Clustering Centers	Points	Recommended Roads
1:00 a.m.–3:00 a.m.	(68,65)	**(115.906,28.6866)**	**109**	**People’s Park North Gate near the road**
		(115.912,28.6862)	102	The intersection of Nanfu Road and Fuzhou Road
		(115.909,28.6865)	90	The intersection of Fuzhou Road and Xian Shi Second Road
7:00 a.m.–9:00 a.m.	(68,62)	**(115.908,28.6584)**	**106**	**Roads near Aier Eye Hospital**
		(115.906,28.6620)	87	Jiangdian District at the exit of Jinggangshan Avenue
		(115.913,28.6638)	78	Nanchang Station West Square Exit
5:00 p.m.–7:00 p.m.	(68,84)	**(115.912,28.8617)**	**121**	**Road section near T2 of Changbei Airport**
		(115.913,28.8582)	36	Changbei Airport Taxi Service Area
	(56,58)	**(115.785,28.6244)**	**434**	**Near Shenzhou Car Rental on Shi Zhongshan Road**
		(115.790, 28.6267)	142	Wuyuan Road and Nanchang West Station intersection section
		(115.786,28.6285)	30	Exit of Nanchang Coach Station West
8:00 p.m.–10:00 p.m.	(62,66)	**(115.845,28.6964)**	**102**	**The intersection of Exhibition Road and Wanda Plaza**
		(115.844,28.6943)	91	Phoenix Center and the intersection with Wanda Plaza
		(115.845,28.6987)	44	The intersection of Fang Hua Road and Sha Jing Road
	(66,64)	**(115.887,28.6788)**	**168**	**Intersection of Parkson Shopping Center and Tianhong Mall**
		(115.892,28.6779)	67	Strongway Building Intersection
		(115.887,28.6752)	64	The intersection of Ruzi Road and Xiangshan South Road

## Data Availability

Data from the 2020 Jiangxi Open Data Innovation Application Competition, in which data generated by cab operations on 9 July 2021 was used.

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
