# Peer review of "Forecasting and Planning Method for Taxi Travel Combining Carbon Emission and Revenue Factors—A Case Study in China"

_ijerph, 2022, doi:10.3390/ijerph191811490_

Round 1

Reviewer 1 Report

The paper is interesting and the authors have good knowledge in the domain. The mathematical model is well presented. The input data are correctly obtained. The results are sustained by the algorithm used in research. I have only one observation. The authors prefer to have references only from Asian authors. This aspect reduces the ability to identify those scientific aspects useful in research. A comparison with other studies from another part of the planet helps to increase the interest in the research carried out by them.

Author Response

Dear Editors and Reviewers:

Thank you for your letter and the reviewers' comments concerning our manuscript entitled “Forecasting and planning method for taxi travel combining carbon emission and revenue factors-A case study in China”. Those comments are all valuable and very helpful for revising and improving our paper, as well as the important guiding significance to our research. We have studied the comments carefully and have made corrections which we hope meet with approval. Revised portions are marked in blue on the paper.

Reviewer 2 Report

Interesting approach to analyzing one road traffic problem. However, the article contains many editorial, descriptive and editorial errors, and therefore requires correction.

The review of the article is difficult due to the lack of line numbering!

In the abstract, please add "a.m." and "p.m." during the research hours. The same issue also occurs in the article, e.g. page 9.

The introduction and conclusions do not describe how the studied phenomenon is influenced by the planning of the network of taxi stands. (p. 2)

Chapter titles should not be the last line on a page (e.g. chapter 1.2.4., 2.2.2.) (p. 2)

It is not described in what software the spatial analysis was carried out (e.g. QGIS, ArcGIS, R, Python or other). Please provide a flowchart describing the data analysis. (p. 4)

No citation of packages / modules / libraries used for spatial analysis. Such algorithms are often open source and proper citation is a form of appreciation for the author's work. (p. 4) Similarly with the algorithms used to analyze the data (p. 7)

The paragraph between Tables 1 and 2 is misaligned. (p. 4)

Some paragraphs have wrong line spacing. (e.g. p. 5)

The source (citation) of Table 4 is not given.

Please explain why formulas (3) - (6) were used for the calculations instead of using projected CRS? (p. 6)

Figure 3 - why are there large water areas in the area of ​​analysis? Do taxis in this area run e.g. on ferries? There is also no indication of the source of the map (looks like OSM). The inscriptions on the map are not in English.

Chart 7 has no designations of the variables on the axes.

Figures 9 and 10. The legend is too small (illegible). I propose to present both drawings on a separate page, in landscape orientation. Figures 11 and 12 are analogous.

Table 7 - The last column is not described in English.

Table 8 - the caption should be above the table.

Good luck, hope to read the revised version soon!

Author Response

(The authors gave the same response as above.)

Reviewer 3 Report

The authors present an implementation of the Forecasting and planning method for taxi travel. The method includes combining carbon emission and revenue factors. The discussed subject is actual within the sustainability transport.

In my opinion: the manuscript does not correspond to the topics of International Journal of Environmental Research and Public Health (and Special Issue IJERPH: "New Theory and Technology of Disaster Monitoring and Prevention", where keywords: safety prevention and monitoring, public security risk, dynamic disaster monitoring and warning, accident prevention and emergency management).

Others – detailed comments - see Annex

Author Response

(The authors gave the same response as above.)

Round 2

Reviewer 3 Report

Dear Authors, Thanks so much for replying to my detailed comments. This manuscript (2ed version) contains many editing errors (Detailed comments - see attachment): - 12-Hour vs. 24-Hour format. A 24-hour clock, sometimes referred to as military time, states the time according to the number of hours that have passed since midnight. Starting at midnight, hours are numbered from 0 to 24, removing the need for designations like am and pm. For example, at 23:00, 23 hours have passed since the beginning of the current day. e.g. 8:00 pm or 20:00 (20:00 pm - error), - fig. 3, 9-14 (resolution, size), - table 6 - 8 (style and 12-Hour vs. 24-Hour format).

Author Response

Dear Editors and Reviewers:

Thank you for your letter and the reviewers' comments concerning our manuscript entitled “Forecasting and planning method for taxi travel combining carbon emission and revenue factors-A case study in China”. Your grasp of the details is very good, and the suggestions you gave are very helpful for us to improve the writing level of the paper. The revised part has been marked with a yellow background
